# Machine Learning Models for Sarcopenia Identification Based on Radiomic Features of Muscles in Computed Tomography

**DOI:** 10.3390/ijerph18168710

**Published:** 2021-08-18

**Authors:** Young Jae Kim

**Affiliations:** Department of Biomedical Engineering, Gachon University, Inchon 21936, Korea; youngjae@gachon.ac.kr; Tel.: +82-32-458-2844; Fax: +82-32-460-2361

**Keywords:** sarcopenia, radiomic feature, machine learning, identification, computed tomography

## Abstract

The diagnosis of sarcopenia requires accurate muscle quantification. As an alternative to manual muscle mass measurement through computed tomography (CT), artificial intelligence can be leveraged for the automation of these measurements. Although generally difficult to identify with the naked eye, the radiomic features in CT images are informative. In this study, the radiomic features were extracted from L3 CT images of the entire muscle area and partial areas of the erector spinae collected from non-small cell lung carcinoma (NSCLC) patients. The first-order statistics and gray-level co-occurrence, gray-level size zone, gray-level run length, neighboring gray-tone difference, and gray-level dependence matrices were the radiomic features analyzed. The identification performances of the following machine learning models were evaluated: logistic regression, support vector machine (SVM), random forest, and extreme gradient boosting (XGB). Sex, coarseness, skewness, and cluster prominence were selected as the relevant features effectively identifying sarcopenia. The XGB model demonstrated the best performance for the entire muscle, whereas the SVM was the worst-performing model. Overall, the models demonstrated improved performance for the entire muscle compared to the erector spinae. Although further validation is required, the radiomic features presented here could become reliable indicators for quantifying the phenomena observed in the muscles of NSCLC patients, thus facilitating the diagnosis of sarcopenia.

## 1. Introduction

Sarcopenia is an illness accompanied by the loss of muscle mass and muscle strength that becomes more prevalent with age. Sarcopenia is closely associated with injury, decreased functioning, and death [1]. With the accelerated aging of the worldwide population in recent years, the prevalence of sarcopenia is increasing. The number of global cases is expected to increase from 50 million in 2010 to 200 million by 2050 [2,3]. Accordingly, sarcopenia is recognized as one of the most serious threats to public health from clinical, social, and economic perspectives [4].

The accurate quantification of muscles is required for the accurate diagnosis of sarcopenia. The quantity and quality of muscles are the main factors to be measured. In particular, muscle quantity is one of the most important indicators for diagnosing sarcopenia. Various methods have been proposed to measure muscle quantity accurately. The most accurate method is to measure the muscle mass manually via computed tomography (CT) or magnetic resonance imaging [5]. However, this approach requires enormous amounts of time and energy. Recently, various attempts at automated measurements of muscle mass have been made using artificial intelligence (AI). In 2020, Blanc-Durand et al. trained a convolutional neural network model using 1025 CT slices to distinguish the muscle areas and reported a dice similarity coefficient (DSC) of 0.97 in 500 test sets that had been separately developed [6]. In the same year, Park et al. trained a fully convolutional network model using 883 L3-level CT slices obtained from 467 patients in an attempt to divide the muscle, subcutaneous fat, and visceral fat and reported a DSC of 0.97 in 426 test sets of 308 patients [7].

In most existing AI-based studies, scholars have attempted to quantify the muscle mass in a prompt and convenient manner. CT images display not only the quantity of muscles, but also provide many other types of information that are difficult to identify with the naked eye [8]. If further quantifiable information about muscles can be acquired from CT images, it will be useful in the diagnosis of sarcopenia. Researchers have recently reported that the CT-derived skeletal muscle radiodensity is closely related to sarcopenia [9,10,11]. In 2020, Ebadi et al. reported that the CT images of hepatocirrhosis patients showed a significant difference (*p* < 0.05) in radiodensity between non-sarcopenia and sarcopenia patients in both men and women [9]. In the case of sarcopenia, such a difference in radiodensity can be indicated by the texture of a muscular cross-section or a structural difference that can also be described by the radiomic features in the radiological area. Radiomic features are obtained from medical images by applying various quantification methods to the features that are difficult to recognize with the naked eye. As quantified information, radiomic features are used to explain the radiological properties of tumors and other diseases [12]. Such radiomic features enable the phenomena present in a muscular cross-section caused by an increase or decrease in the muscle to be quantified and digitized into various radiological indicators. If radiomics techniques can respond sensitively to the phenomena present on a muscular cross-section, they can be used to identify sarcopenia through classifiers such as machine learning.

In this study, we aimed to find radiomic features that are useful for identifying sarcopenia in CT images of non-small cell lung cancer (NSCLC) patients and to verify the performance of machine learning in identifying radiomic features for the diagnosis of sarcopenia.

## 2. Materials and Methods

### 2.1. Ethics Statement

This retrospective study was approved by the Institutional Review Board (IRB) of Gachon University Gil Medical Center, and the requirement of the informed consent of the patients was waived (GCIRB 2020-251, 30/06/2020).

### 2.2. Data Collection

We retrospectively collected the CT data of 247 patients diagnosed with pathologically proven NSCLC at the Gil Medical Center of Gachon University between January 2011 and December 2016. To eliminate the effects of external factors as much as possible, we limited the participants of this study to NSCLC patients. The patients included 149 men (60.32%) and 98 women (39.68%). The average age of the male patients was 63 ± 9.26 years (range: 36–81 years), and the average age of the female participants was 62 ± 10.38 years (range: 33–80 years).

To train the machine learning model, we divided the datasets into training data and test data. For use as the test data, 20% of the data were randomly selected from the sarcopenia group and the non-sarcopenia group separately (sarcopenia, *n* = 12; non-sarcopenia, *n* = 37). The remaining data were used as the training data (sarcopenia, *n* = 49; non-sarcopenia, *n* = 149). Figure 1 provides a flowchart of the data collection and analysis procedures.

### 2.3. CT Examinations

The patients underwent contrast-enhanced multi-detector CT scans (Somatom Definition 64, and Somatom Definition Flash, Siemens Medical Solutions, Erlangen, Germany). A contrast agent was injected at a volume of 2 mL/kg of body weight (maximum 150 mL) through an 18-gauge peripheral venous access device at a flow rate of 4 mL/s and portal venous imaging was obtained one minute after achieving a 50 HU enhancement of the descending aorta.

### 2.4. Definition of Sarcopenia

To define sarcopenia, we quantified the muscular areas using an in-house-developed software program (Gachon_DeepBody, developed in the GCUMC, Incheon, Korea) and the collected L3 CT images. Gachon_DeepBody automatically extracts the images of muscles, subcutaneous fat, and visceral fat through a trained U-Net-based deep learning model and provides the areas of each body component [13] (see Figure 2). Using the software program, we automatically identified the muscles in the CT images and secured the muscular areas at the L3 level by manually correcting the erroneously identified areas. The identified areas were also reviewed and corrected by a radiologist with 13 years of experience.

We calculated the L3 muscle indices (L3MI, cm^2^/m^2^) by normalizing the cross-sectional area of the entire muscle for height at the L3 level. Sarcopenia was defined as an L3MI of less than 55 cm^2^/m^2^ for men and less than 39 cm^2^/m^2^ for women, as proposed by the International Consensus for Cancer Cachexia [14]. Based on each cut-off, the presence of sarcopenia was determined in all data, which were used as ground truth data after a final review by the radiologist.

### 2.5. Regions of Interest (ROIs) on Muscle

ROIs for two types of muscle were also defined to perform comparisons of the muscles, from which radiomic features were to be extracted. The first ROI was defined as the overall muscle area, and the second was defined as a part of the erector spinae muscle.

The muscle areas obtained from the L3MI calculation process were used for the ROI of the overall muscle area. ImageJ (version 1.53e), opensource software provided by the National Institutes of Health (NIH, Bethesda, MD, USA), was used to define the ROI of the partial areas of the erector spinae muscle. Box-shaped ROIs with dimensions of 1.5 cm × 1.5 cm were drawn in the left and right erector spinae muscles, respectively, on an L3 level CT slice. Two ROIs were collected from each patient. Figure 3 provides examples of the ROIs that were collected.

### 2.6. Radiomic Feature Extraction

Using the CT images, we extracted radiomic features from the CT values in each ROI, either for the entire muscle area or for the partial areas of the erector spinae muscle. A Python package called Pyradiomics (version 3.6.2, https://github.com/Radiomics/pyradiomics.git, (accessed on 1 July 2021) was used to extract these features [15]. Pyradiomics is an open-source platform for extracting radiomic features from medical images. We used Pyradiomics to analyze the following six types of radiomic features, excluding morphological features: first-order statistics, gray-level co-occurrence matrix (GLCM), gray-level size zone matrix (GLSZM), gray-level run length matrix (GLRLM), neighboring gray-tone difference matrix (NGTDM), and gray-level dependence matrix (GLDM) [16].

The first-order statistics indicate the distribution of pixel intensities within an area [17]. The GLCM explains a joint probability function by representing the number of combinations of two neighboring pixels in an area through a matrix [18]. The GLSZM quantifies the number of connected pixels in a matrix that share the same gray-level intensity in an area [19]. The GLRLM quantifies the lengths of successive pixels in a matrix that have the same gray-level value in an area [20]. The NGTDM quantifies the difference between adjacent gray-level values within a specified distance through a matrix [21]. The GLDM quantifies the dependence of gray-levels in an area through a matrix [22]. The dependence of the gray-level is defined as the number of pixels connected within a particular distance and is dependent on the center pixel. In this study, we extracted 94 radiomic features: 19 first-order statistics features, 24 GLCM features, 16 GLSZM features, 16 GLRLM features, 5 NGTDM features, and 14 GLDM features. As different cut-offs are used to diagnose sarcopenia for different sexes, we added the variable of sex, yielding a total of 95 radiomic features.

### 2.7. Feature Selection

Of the 95 features, those with variance inflation factors (VIFs) of 10 or above were considered to be multicollinear and were removed. The permutation feature importance algorithm was implemented for the remaining variables to derive the importance of each feature. This algorithm calculates the importance by determining the change in a particular score, whereas the index of each feature is randomly shuffled in a model that has been initially trained once [23,24]. As illustrated in Figure 4, four features (sex, coarseness, skewness, and cluster prominence) were selected.

### 2.8. Machine Learning Model to Identify Sarcopenia

We trained machine learning models to distinguish between sarcopenia and non-sarcopenia using the already-constructed training data. Four types of machine learning models were trained for comparisons: logistic regression (LR), support vector machine (SVM), random forest (RF), and extreme gradient boosting (XGB).

LR is a statistical method for predicting the probability using a linear combination of independent variables. LR classifies values by applying a logistic function to a coefficient calculated via linear regression [25]. The SVM technique is a classification model for determining a hyperplane in which the margin or distance between each data group and the baseline for classifying the data is maximized [26]. As an ensemble model with an expanded form of the decision tree method, the RF approach develops multiple decision trees and determines a result with the optimal performance based on the votes for the classification of each tree [27]. XGB was developed to overcome the disadvantages of gradient boosting. The execution of this algorithm is rapid, and it demonstrates excellent prediction performance. It also includes the function of overfitting regularization, which performs internal cross-validation in each iterative execution [28].

We selected the optimal parameters for each machine learning model through a grid search [29] (see Table 1). To prevent data imbalances, we also performed oversampling of the training data for the sarcopenia group by implementing a synthetic minority oversampling technique (SMOTE) algorithm and subsequently trained the machine learning models [30].

### 2.9. Statistical Analysis

The identification performance of each machine learning model (LR, SVM, RF, and XGB) was evaluated in terms of the accuracy, sensitivity, specificity, positive predictive value (PPV), and negative predictive value (NPV), which were calculated based on the true positive (TP), false positive (FP), false negative (FN), and true negative (TN). To evaluate the predictive performance of each machine learning model, we calculated the area under the curve (AUC) of the receiver operating characteristic (ROC) curve. The Delong method [31], based on non-parametric statistics, was used to compare the AUC of the ROC curves, whereas the Bonferroni method was used to correct for statistical multiplicity. 

The machine learning models and their identification performances were evaluated using the scikit-learn library (version 0.23.2). The statistical analysis of the ROC was performed using MedCalc (version 14.0, MedCalc Software Ltd., Mariakerke, Belgium). Clinical data were statistically analyzed using SPSS (version 20, IBM Corp., Armonk, NY, USA). Statistical significance was set at *p* < 0.05.

## 3. Results

The basic characteristics of the patients in the sarcopenia and non-sarcopenia groups were collected, as shown in Table 2.

In this study, we selected four features and trained each machine learning model using them. Next, we compared the sarcopenia identification performances of these models based on the separately constructed testing data. Five-fold cross-validation was employed because the test data were not sufficient. Training and validation were iterated five times, and every dataset was used for validation once. Table 3 and Figure 5 present the results of the cross-validation for the sarcopenia identification performance of each machine learning model.

Cross-validation revealed that the XGB model exhibited the best performance for the entire muscle. The XGB model exhibited evenly distributed sensitivity (0.771, CI: 0.645–0.869) and specificity (0.839, CI: 0.778–0.889), and the highest accuracy (0.822, CI: 0.768–0.868). On the other hand, the SVM model exhibited the worst performance. The accuracy of SVM (0.790, CI: 0.7332–0.839) was lower than that of LR (0.777, CI: 0.720–0.828). However, LR demonstrated evenly distributed sensitivity (0.803, CI: 0.682–0.894) and specificity (0.769, CI: 0.702–0.827), whereas SVM exhibited biased results between sensitivity (0.377, CI: 0.256–0.510) and specificity (0.925, CI: 0.877–0.958). In the case of the erector spinae muscle, the LR model demonstrated the best performance. Although the accuracy of the RF model (0.692, CI: 0.631–0.749) was slightly higher than that of the LR model (0.664, CI: 0.601–0.723), the LR model exhibited superior performance in terms of the deviation between sensitivity and specificity. The SVM model exhibited the worst performance. The SVM model exhibited a large deviation between sensitivity (0.131, CI: 0.058–0.242) and specificity (0.909, CI: 0.858–0.946), and the lowest accuracy (0.556, CI: 0.492–0.619). In the comparison between the entire muscle and the erector spinae muscle, all models exhibited higher accuracy, sensitivity, specificity, PPV, and NPV for the entire muscle than for the erector spinae muscle.

In the comparative analysis of the AUC between the models, the XGB model exhibited a statistically significant difference compared to SVM (*p* = 0.0008) and RF (*p* = 0.0245), respectively, but it showed no significant difference with LR (*p* = 0.0533). The ROC analysis of the erector spinae muscle achieved the highest performance with the LR model (0.750, CI: 0.691–0.802). In the comparative analysis of the AUC between the models, the LR model exhibited a statistically significant difference compared to SVM (*p* = 0.0001) and XGB (*p* = 0.0331), respectively, but it showed no significant difference with RF (*p* = 0.0696). The AUC of each model was found to be greater for the entire muscle than for the erector spinae muscle. Additionally, SVM (*p* = 0.0005), RF (*p* = 0.0037), and XGB (*p* = 0.0001) indicated a statistically significant difference in AUC between the entire muscle and the erector spinae muscle, but the results of LR (*p* = 0.0716) were not significant different.

Figure 6 shows a heatmap of the effects of the four features on the identification of sarcopenia in each machine learning model. In all the models, the most influential parameter for the identification of sarcopenia was sex, with heatmap values of 0.363 for LR, 0.220 for SVM, 0.279 for RF, and 0.338 for XGB. Coarseness was the most influential parameter among the radiomic features, with heatmap values of 0.039 for LR, 0.087 for SVM, 0.056 for RF, and 0.111 for XGB.

## 4. Discussion

In this study, we extracted a large number of radiomic features from the L3 level CT slices of 247 patients diagnosed with NSCLC and subsequently selected the relevant features for the effective identification of sarcopenia. Furthermore, we validated the performance of these features through machine learning, focusing on how appropriately they could identify sarcopenia and non-sarcopenia.

The four features of sex, coarseness, skewness, and cluster prominence were selected as the relevant features for effectively identifying sarcopenia. Among these features, sex was the most important feature because it resulted in the use of different muscle index cut-offs in diagnosing sarcopenia. Coarseness is one of the NGTDM features and indicates the average difference between a center pixel and its neighboring pixel in NGTDM, as well as the spatial change rate. The higher the coarseness, the lower the spatial change rate and the more uniform the local texture. Skewness is a first-order feature and suggests an asymmetrical distribution of values with respect to the average value in a histogram. The lower the skewness, the more skewed the histogram is to the left. The higher the skewness, the more skewed the histogram is to the right. On the one hand, the left-side skewness of a histogram indicates that the overall distribution of the CT values is low. On the other hand, right-side skewness signifies a high distribution of CT values. Cluster prominence is a GLCM feature that indicates the asymmetry of the GLCM. The higher the cluster prominence, the larger the asymmetry with respect to the average. Furthermore, the smaller the cluster prominence, the closer the peak to the average value and the smaller the deviation from the average value. In GLCM, the large asymmetry with respect to the average implies a large difference between the neighboring pixels, indicating a rough surface texture of the muscle.

The features selected in this study had the following magnitudes for the sarcopenia and non-sarcopenia groups: 0.00022 ± 0.00013 and 0.00020 ± 0.00012 (*p* = 0.188) for coarseness, −0.372 ± 0.304 and −0.328 ± 0.421 (*p* = 0.369) for skewness, and 15.517 ± 25.330 and 11.578 ± 5.012 (*p* = 0.232) for cluster prominence, respectively. The sarcopenia group showed a slightly larger coarseness, smaller skewness, and larger cluster prominence than the non-sarcopenia group. Considering these characteristics in their entirety, the cross-section of the sarcopenia muscle displayed more low-intensity values than the non-sarcopenia muscle. Therefore, the cross-section of the sarcopenia muscle is more likely to have a rough surface. Based on these results, it can be inferred that, as the muscle density was lower in the sarcopenia group, fat intrusion occurred and the low Hounsfield unit of fat influenced the intensity distribution and texture of muscle in the CT images.

Considering the permutation importance of each feature, the optimal identification performance for sarcopenia can be ensured by considering the four features collectively. In other words, sarcopenia needs to be explained by comprehensively considering all four features. However, because the individual radiomic features do not exhibit any statistically significant differences between the sarcopenia and non-sarcopenia groups, an individual feature is limited in its ability to explain sarcopenia. Accordingly, univariate analysis needs to be conducted in the future to examine the correlations between the individual radiomic features and sarcopenia.

Sex is the largest contributing factor in the identification of sarcopenia. However, the data used in this study are susceptible to sex-specific bias. Only seven out of 61 participant sarcopenia patients were female. Additionally, there were only seven patients with sarcopenia in the data based on all 104 female patients. This imbalance in the data may lead to biased learning and is a limitation in this experiment. Therefore, it is necessary to conduct further experiments in the future by minimizing the issue of sex imbalance through additional data collection.

Among the four machine learning models of LR, SVM, RF, and XGB, the XGB model demonstrated the best performance for the identification of sarcopenia for the entire muscle. The XGB model showed an AUC of 0.837 (CI: 0.758–0.881). The other three models demonstrated satisfactory performance. In the case of the SVM approach, the AUC was 0.778 (CI: 0.721–0.828), corresponding to appropriate performance; however, the sensitivity of 0.377 (CI: 0.256–0.510) and specificity of 0.925 (CI: 0.877–0.958) indicated biased results. To prevent data imbalance between the sarcopenia and non-sarcopenia groups as much as possible, we conducted oversampling by applying a SMOTE technique to the training process. Nevertheless, biased training could not be prevented in a few models. Furthermore, it was not possible to perform external validation in this study, which is essential in order to identify biased training and model overfitting. Further research is required to validate and solve the problems of data imbalance and overfitting by collecting additional data for external validation.

We conducted a comparative experiment to determine whether the entire muscle could be represented by the area of a partial muscle. If the area of a partial muscle can exhibit performance similar to that of the entire muscle, it would be possible to avoid spending the time and energy necessary to acquire the entire muscle area from CT. Accordingly, we set box-shaped regions with dimensions of 1.5 cm × 1.5 cm in the erector spinae muscle and conducted a comparative experiment on the area of the entire muscle. However, the identification performance of machine learning for the partial region of the erector spinae muscle was inferior to that of the entire muscle. The LR, SVM, RF, and XGB models all showed low performance, and the LR model, which had the best performance, had an AUC of 0.750 (CI: 0.691–0.802). This finding may support the conclusion that the partial region set in the erector spinae muscle is too small to represent the change in the entire muscle. However, this result does not imply that the partial region of the erector spinae muscle cannot be used to identify sarcopenia. Generally, diagnostic accuracy is considered to be good when the AUC is 0.7 or greater. Therefore, we believe that the partial region of the erector spinae muscle has sufficient potential to be used in the identification of sarcopenia. Further verification is necessary in the future through various comparative studies that consider the location of partial regions and measurement ranges.

There are several limitations associated with the data used in this study. First, because NSCLC patients were the only participants in this study, we cannot conclude that the results of the study are applicable to other cases. Further experiments and validations are necessary with respect to various ethnic groups, sexes, and symptoms. A comparative experiment also needs to be conducted for various modalities, such as DXA. Second, the images used in this study were collected from a single center, but taken with two pieces of equipment. Although the difference in quality due to the equipment may not be sufficiently significant to alter the direction of the results, we cannot rule out the possible influence of the equipment. This is an issue that occurs in all radiomics studies. To address the bias issue created by the use of different equipment, the models need to be generalized by collecting sufficient data, which is very difficult to achieve. Further research is needed to understand the extent of the influence of different equipment on the quality of images. Third, sex is the largest contributing factor in the identification of sarcopenia. However, the data used in this study are susceptible to sex-specific bias. Only seven out of 61 participant sarcopenia patients were female. There were only seven patients with sarcopenia in the data based on all 104 female patients. This imbalance in the data may lead to biased learning and is a limitation in this experiment. Therefore, it is necessary to conduct further experiments in the future by minimizing the issue of sex imbalance through additional data collection.

Although some limitations are yet to be addressed, in this study, the radiomic features in muscles exhibited the potential to serve as effective indicators of sarcopenia and non-sarcopenia through machine learning. If sufficient validation is ensured by performing various additional experiments, then these radiomic features could become reliable indicators explaining the phenomena appearing on the cross-sections of muscles and could be used to diagnose sarcopenia.

## 5. Conclusions

In conclusion, radiomic features can be used to quantify the phenomena observed in the muscles of NSCLC patients and have the potential to become useful radiologic indicators for the diagnosis of sarcopenia. If sufficient validation is ensured by further research, these radiomic features could be used as indicators providing diverse information about muscles that are generally not visible to the naked eye, as well as the mass, density, and strength of muscle, and facilitating accurate sarcopenia diagnoses.

## Figures and Tables

**Figure 1 ijerph-18-08710-f001:**
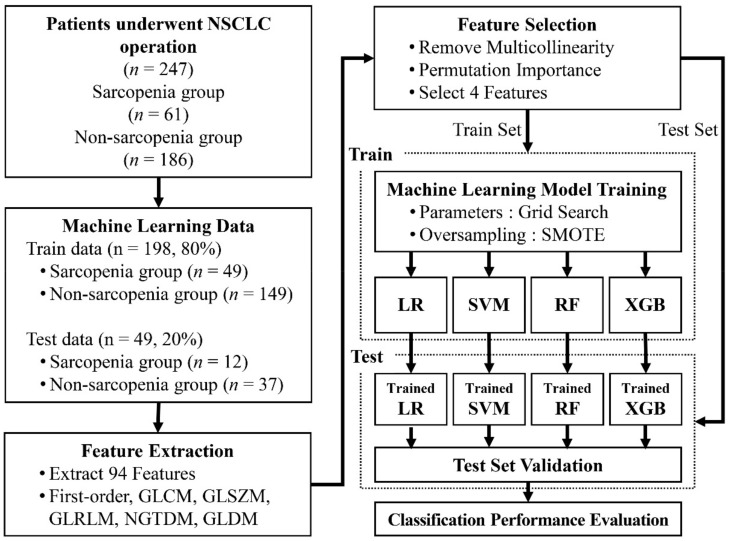
Flowchart of sarcopenia data collection and analysis. NSCLC, non-small cell lung carcinoma; GLCM, gray-level co-occurrence matrix; GLSZM, gray-level size zone matrix; GLRLM, gray-level run length matrix; NGTDM, neighboring gray-tone difference matrix; GLDM, gray-level dependence matrix; SMOTE, synthetic minority oversampling technique; LR, logistic regression; SVM, support vector machine; RF, random forest; XGB, extreme gradient boosting.

**Figure 2 ijerph-18-08710-f002:**
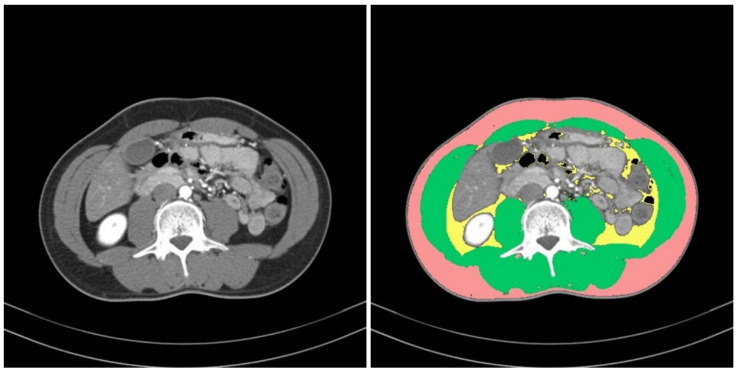
Areas of body components identified by Gachon_DeepBody: muscle (**green**), subcutaneous fat (**red**), visceral fat (**yellow**).

**Figure 3 ijerph-18-08710-f003:**
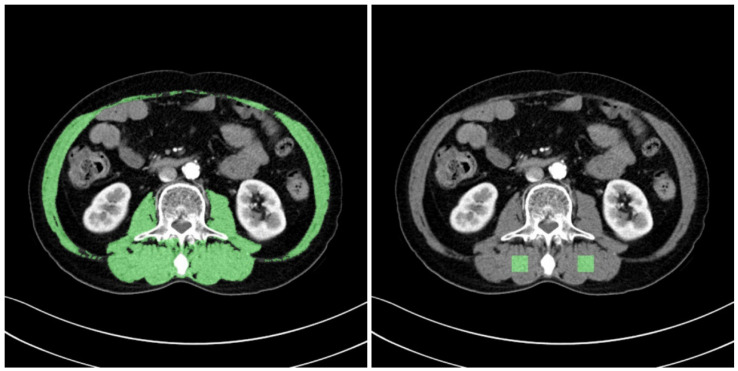
Examples of ROIs for two types of muscles. ROI for the entire muscle area (**left**), ROI for the left and right erector spinae muscles (**right**).

**Figure 4 ijerph-18-08710-f004:**
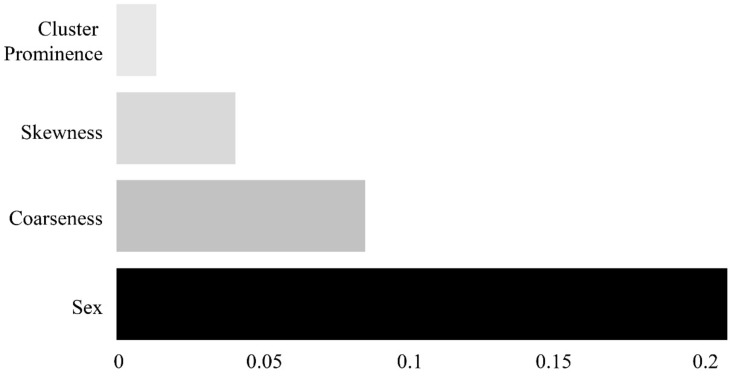
Feature importance plot showing the relative importance of four features with respect to the identification of sarcopenia and non-sarcopenia groups.

**Figure 5 ijerph-18-08710-f005:**
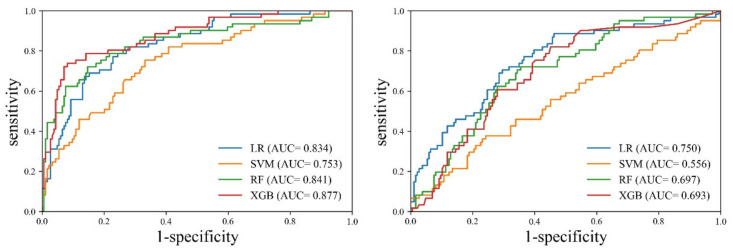
Comparison of ROC curves among different machine learning models for the identification of sarcopenia. The left graph presents the ROC curves of the entire muscle, where XGB has the highest AUC. The right graph shows those of the erect spinae muscle, where LR has the highest AUC. LR, logistic regression; SVM, support vector machine; RF, random forest; XGB, extreme gradient boosting; ROC, receiver operating characteristic; AUC, area under the curve.

**Figure 6 ijerph-18-08710-f006:**
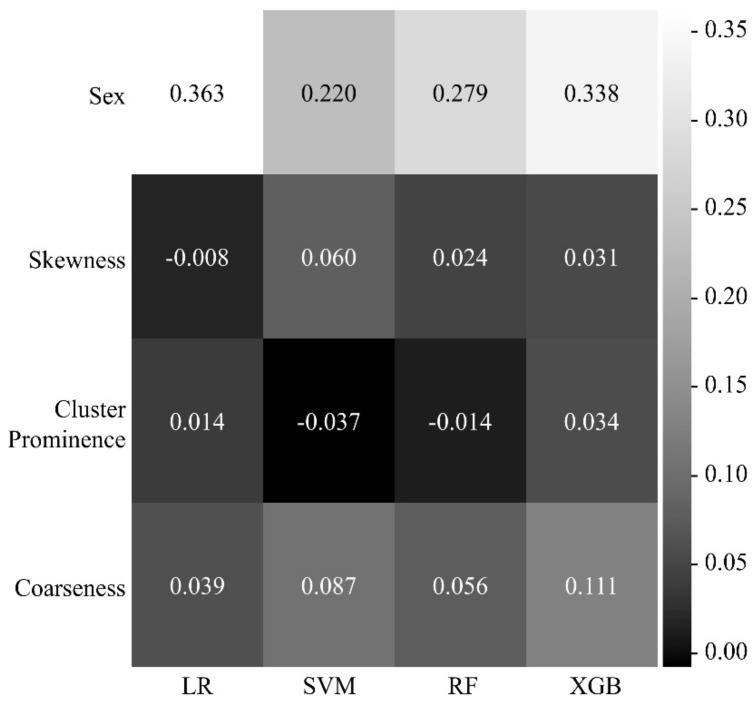
Heatmap of the effects of the functions selected in each machine learning model for the identification of sarcopenia. The higher the heatmap value (that is, the closer the color is to white), the more intense the impact of a feature on the identification of sarcopenia. LR, logistic regression; SVM, support vector machine; RF, random forest; XGB, extreme gradient boosting.

**Table 1 ijerph-18-08710-t001:** Parameters used in the four types of machine learning models.

Model	Parameters
LR	C: 7.079, penalty: 12, random state: 1818
RF	max depth: 27, min samples split: 7, n estimators: 80, random state: 1818
SVM	C: 141.3094, gamma: 25.7714, probability: True, random state: 1818
XGB	max depth: 2, learning rate: 0.0093, n estimators: 979, random state: 1818

LR, logistic regression; SVM, support vector machine; RF, random forest; XGB, extreme gradient boosting.

**Table 2 ijerph-18-08710-t002:** Characteristics of the patients in the sarcopenia and non-sarcopenia groups and comparisons between the groups.

Characteristics	Number of Patients (%)		*p* Value
	Sarcopenia (*n* = 61)	Non-Sarcopenia (*n* = 186)	
Age (years)	63.508 ± 9.230	62.263 ± 9.790	0.3831
Sex			<0.0001
Males	54(88.5%)	89(47.8%)	
Females	7(11.5%)	97(52.2%)	
Height (cm)	1.655 ± 0.078	1.598 ± 0.085	<0.0001
Weight (kg)	58.557 ± 9.368	64.237 ± 10.668	0.0003
BMI (kg/m^2^)	21.324 ± 2.745	25.089 ± 3.196	<0.0001
L3 muscle index (cm/m^2^)	129.975 ± 20.979	148.538 ± 37.137	<0.0001
L3 VAT area (cm/m^2^)	100.661 ± 58.589	132.677 ± 65.647	0.0008
L3 SAT area (cm/m^2^)	99.180 ± 57.466	153.101 ± 68.319	<0.0001
Histology (%)			0.0358
ADC	32(53.3%)	129(69.0%)	
SCC	24(40.0%)	43(23.0%)	
Others	4(6.7%)	15(8.0%)	
Neoadjuvant therapy (%)	5(8.3%)	11(5.9%)	0.7116
Operation type (%)			0.4267
Lobectomy	51(85.0%)	168(89.8%)	
Others	9(15.0%)	19(10.2%)	
Pathologic stage (%)			0.1619
IA	17(28.3%)	66(35.3%)	
IB	16(26.7%)	51(27.3%)	
IIA	13(21.7%)	26(13.9%)	
IIB	4(6.7%)	4(2.1%)	
IIIA	8(13.3%)	38(20.3%)	
IV	2(3.3%)	2(1.1%)	

Values are the mean ± standard deviation or counts (proportions). *p*-value: Student *t*-test. BMI, body mass index; VAT, visceral adipose tissue; SAT, subcutaneous adipose tissue; ADC, adenocarcinoma; and SCC, squamous cell carcinoma.

**Table 3 ijerph-18-08710-t003:** AUC, sensitivity, specificity, PPV, and NPV of each machine learning model for the identification of sarcopenia in the entire muscle and the erector spinae muscle.

	AUC (95% CI)	Accuracy (95% CI)	Sensitivity (95% CI)	Specificity (95% CI)	PPV (95% CI)	NPV (95% CI)
(Entire muscle)						
LR	0.832 (0.779–0.876)	0.777 (0.720–0.828)	0.803 (0.682–0.894)	0.769 (0.702–0.827)	0.533 (0.460–0.604)	0.923 (0.877–0.952)
SVM	0.778 (0.721–0.828)	0.790 (0.733–0.839)	0.377 (0.256–0.510)	0.925 (0.877–0.958)	0.622 (0.475–0.749)	0.819 (0.788–0.847)
RF	0.828 (0.775–0.873)	0.798 (0.742–0.846)	0.721 (0.592–0.829)	0.823 (0.760–0.875)	0.571 (0.485–0.653)	0.900 (0.857–0.931)
XGB	0.837 (0.758–0.881)	0.822 (0.768–0.868)	0.771 (0.645–0.869)	0.839 (0.778–0.889)	0.610 (0.523–0.691)	0.918 (0.875–0.947)
(Erector spinae muscle)						
LR	0.750 (0.691–0.802)	0.664 (0.601–0.723)	0.771 (0.645–0.869)	0.629 (0.555–0.699)	0.405 (0.351–0.462)	0.893 (0.839–0.931)
SVM	0.556 (0.492–0.619)	0.717 (0.656–0.772)	0.131 (0.058–0.242)	0.909 (0.858–0.946)	0.320 (0.176–0.509)	0.761 (0.741–0.780)
RF	0.697 (0.636–0.754)	0.692 (0.631–0.749)	0.590 (0.457–0.715)	0.726 (0.656–0.789)	0.414 (0.340–0.491)	0.844 (0.798–0.881)
XGB	0.693 (0.631–0.750)	0.656 (0.593–0.715)	0.607 (0.473–0.729)	0.672 (0.600–0.739)	0.378 (0.313–0.447)	0.839 (0.790–0.878)

LR, logistic regression; SVM, support vector machine; RF, random forest; XGB, extreme gradient boosting; AUC, area under the curve; PPV, positive predictive value; NPV, negative predictive value; CI, confidence interval.

## Data Availability

The datasets generated or analyzed during the current study are available from the corresponding author upon reasonable request.

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
