# Peer review of "Machine Learning Models for Sarcopenia Identification Based on Radiomic Features of Muscles in Computed Tomography"

_ijerph, 2021, doi:10.3390/ijerph18168710_

Round 1

Reviewer 1 Report

In general, the manuscript is well written. Here are some minor comments below.

  1. The study aim should be clearly state in 1-2 sentences in the last sentence of the last paragraph. The expression of the original last paragraph can eb re-written to address the point.
  2. A brief clinical data of those patients with NSCLC should be provided. For example, cancer staging, duration between proved diagnosis and CT scan, whether they were under treatment or not?
  3. How did you obtain the ‘ground truth’? Did all annotation/marked areas of body components in all images check by experienced radiologists?
  4. Sex is the strongest contributing factor, but only 7 women as compared to 54 men participated in sarcopenia group? This potential bias should be mentioned in the Limitation section.
  5. Please remove all data from the test results (line 219-232) because these data have been well presented on Table 3. In the test, just described interpretation of the results.
  6. Similarly, authors can use a Table to show data from line 233-240, and described key interpretation in the text in short.

Reviewer 2 Report

Thank you very much for the opportunity to review your paper. It is very interesting, but I think there are a few problems with the paper.

1-1. The authors created several models (LR, SVM, RF, and XGB). They then compared their results, but MedCalc's built-in method of comparing ROC curves should only allow for comparisons between two groups. It is unclear in which group there was a difference, so please provide details.

1-2. Also, please describe the statistical method used to compare the ROC curves.

1-3. There may be a problem with statistical multiplicity, so please mention how to avoid it.

2-1. The authors described that the machine learning models tended to exhibit improved performance for the entire muscle, than for the erector spinae muscle (lines 231-232, page 8). Is this proven by statistics?

2-2. In addition, for the diagnosis of Sarcopenia, measurements on the entire muscle are standard, while measurements on the erector spinae muscle are non-standard. I do not think it is necessary to add the measurement on erector spinae muscle, which is inferior in performance, in the paper.

  1. Referring to Table 2, the L3 SAT area shows the largest difference between Sarcopenia and Non-sarcopenia. Shouldn't we use fat mass as an indicator rather than muscle mass?

  1. Is it possible that the measurement of radiomic features in muscles is affected by differences in image quality due to the equipment used to take the images? This is a single-center study, but is there a single CT system or multiple systems?

  1. The authors described ‘Based on these results, it can be inferred that, as the muscle density was lower in the sarcopenia group, fat intrusion occurred and the low Hounsfield unit of fat influenced the intensity distribution and texture of muscle in the CT images (lines 284-286, page 10)’. Looking at Figures 2 and 3, we can see that contrast media was used. If the authors are going to evaluate the CT values of the muscles, they need to include the use of contrast media in the methods section. Were contrast agents used in all cases, and were the images taken at the same contrast timing? If some patients used contrast media and some did not, I believe that the bias in the sarcopenia and non-sarcopenia groups needs to be considered.

Round 2

Reviewer 2 Report

Thank you for revising your manuscript.
I think the authors have responded appropriately to the comments I have noted.
I have no specific comments on the revised manuscript.